# Implementation of a Full Digital Workflow by 3D Printing Intraoral Splints Used in Dental Education: An Exploratory Observational Study with Respect to Students’ Experiences

**DOI:** 10.3390/dj11010005

**Published:** 2022-12-26

**Authors:** Pablo Kraemer-Fernandez, Sebastian Spintzyk, Ebru Wahl, Fabian Huettig, Andrea Klink

**Affiliations:** 1Department of Prosthodontics at University Clinic for Dentistry, Oral Medicine, and Maxillofacial Surgery with Dental School, Tuebingen University Hospital, Osianderstr. 2-8, 72076 Tuebingen, Germany; 2Section Medical Materials Science and Technology, University Hospital Tuebingen, Osianderstr. 2-8, 72076 Tuebingen, Germany; 3ADMiRE Research Center—Additive Manufacturing, Intelligent Robotics, Sensors and Engineering, School of Engineering and IT, Carinthia University of Applied Sciences, Europastraße 4, 9524 Villach, Austria

**Keywords:** dentistry, undergraduate medical education, CAD/CAM, questionnaires, dental students, additive manufacturing, oral splints, intraoral scanning, vat photopolymerisation

## Abstract

Fully digital workflows gained acceptance in dental practice and thereby are of interest for undergraduate education. An exploratory clinical observation was designed to track the implementation of such a workflow with novice digital users in order to describe its feasibility, time investment, and pitfalls. Methods: Students were invited to provide feedback for their experiences with a training module that consisted of the following: intraoral scanning, computer-aided design (CAD), manual finishing, and insertion of a 3D-printed bite splint for the lower jaw. Results: A total of 82 fourth-year students participated in the module. The average time required to perform an intraoral scan was 17 m 5 s, and all students were able to design a splint with an average time of 2 h 38 m. Students who indicated prior experience with CAD seem to outperform inexperienced students in both CAD task completion and intraoral scanning. The initial fit was reported as clinically acceptable by 68.5% of the participants, while 79% rated the workflow as very good to satisfactory and indicated that the training was helpful for dental practice. Conclusions: The implementation of a digital workflow in undergraduate dental education is feasible and has acceptable clinical results. However, CAD is time-intensive, and the experience can be challenging.

## 1. Introduction

Digitalization in dentistry changed several dental treatments in recent years [1,2]. Solutions for completely digital processes were facilitated by combining intraoral scanners (IOS), computer-aided design (CAD) software, and computer-aided manufacturing (CAM) [3,4]. While only single-tooth restorations were a reliable option in the beginning [5], the limits of complete digital workflows in dentistry have almost been overcome [6,7,8,9]. Lately, the precision and trueness of complete jaw impressions taken with intraoral scanners have caught up considerably to conventional impressions [10,11,12]. In addition to being able to capture a high local resolution image of a single tooth, intraoral scanners have also improved their dimensional accuracy when capturing complete dentate jaws [13,14,15,16,17,18].

In response to this comparable development within dental practices in most countries, dental education must catch up [19].

Dental schools report the integration of IOS predominantly to fabricate single-tooth restorations in a patient setting or on mannequin head-mounted models [20,21,22,23,24]. In order to obtain reliable full-arch scans, the scan path must be adapted to the narrowness of a dentate oral cavity, the dimensions of the scanning tip, and its limited field of view [25,26,27,28]. Therefore, the operator’s experience can significantly impact the outcome of the scanning process [29]. However, it is not entirely clear whether experience, which can also be represented in a learning curve, correlates with accuracy or whether the experience is rather an increase in efficiency in capture operations [30,31].

Concerning dental laboratories, subtractive manufacturing has been complemented and partially substituted with additive manufacturing (AM) during the past five years [32,33,34]. Besides the current focus on the digital workflow with the additive manufacturing of complete dentures and fixed dental prostheses [35,36], the production of additively manufactured bite splints has already been in the spotlight for several years [37]. AM-based splint fabrication represents a cost-efficient alternative to subtractive manufacturing due to its requirements with respect to dimension, durability or wear, and the ease of applying the process via unfilled clear acrylics [37,38,39,40]. The successful AM of occlusal splints from digital impressions was shown in several clinical and in vitro studies performed by experienced users [37,38,41].

Thus, whether dental students with only “analog” experiences are able to control digital workflows consisting of intraoral scanning and CAD design in a dentate in vivo setting remains questionable. This information is crucial for implementing full-arch IOS and CAD/CAM within undergraduate dental education, and it corresponds with the time investments that must be considered within the curriculum.

Therefore, a clinical observational study should explore the experiences and results of novice users (e.g., students) performing a fully digital workflow to fabricate mandibular occlusal splints by means of AM based on intraoral scans and CAD.

With regard to the exploratory character, no hypothesis was set.

## 2. Materials and Methods

### 2.1. Study Population

Throughout four terms (October 2017–July 2019), dental students in their 4th year of undergraduate dental studies were invited to participate in the teaching module “digital workflow” and to provide feedback for their experiences in a structured accompanying evaluation. At that time, the students were already preclinically educated and practically trained in the conventional production of bite splints, and they had at least 15 days of experience with respect to applying the active clinical treatment of patients under supervision. However, their dental studies did not include practical training with respect to digital workflows: neither CAD nor IOS.

This prospective data collection within an educational setting was reviewed and allowed by the Institutional Ethical Review Board (047/2018BO2). The participants were enrolled based on their written informed consent after being informed with information about the aims and objectives of the evaluation. With regard to ethical standards, their participation and performance had no influence on their study credits, and they were allowed to end their participation at any time without providing reasons.

### 2.2. Study Design

The teaching module under investigation consisted of four tasks: intraoral scan, digital design, finalization, and the insertion of a splint for the lower jaw. All tasks must be performed by each participant. In order to avoid patient bias and to ensure a level of comparability, each active participant (ATP) needed a passive tandem partner (PTP) in order to perform the workflow (see Figure 1).

### 2.3. Questionnaires for the Evaluation of the Tasks

The questionnaires were developed based on a literature review about students’ perspectives using CAD/CAM in dental education and their own experiences from using full digital workflows [23,38,42,43,44,45]. Students, dental technicians, and dentists peer-reviewed the questions to ensure understandability and relevance. The paper-and-pencil evaluation forms were used specifically for each task. The questionnaires encompassed single- and multiple-choice questions, unlabeled visual assessment scales (VAS) of 100 mm with two denoted poles, and free text options (Appendix A). The questionnaires for Task 1 (intraoral scan) and Task 4 (insertion of the splints) were separated for the active tandem partner (ATP) performing the treatment and for the passive tandem partner (PTP) receiving the scan and finalized splint. The questionnaires were available in German and English (Appendix A). They each contained 49 questions, of which 44 questions were related with coping with the difficulty and perception of the tasks, and 1 question was related to previous personal experiences with CAD in general.

### 2.4. Task 1: Intraoral Scanning

At the beginning of each course, all students participating in the module received an oral presentation with a live demonstration of a complete jaw scan on a volunteer (Trios 3 color, see Table 1) by the study’s supervisor (PKF). Bite-scanning was performed with a previously registered and inserted wax plate (Beauty-Pink extra hard 3 mm, Miltex GmbH, Rieheim-Weilheim, Germany) to adjust the jaw’s relation approximately to the desired thickness of the splint. All information was made available to the students. The tandem partners scheduled their intraoral scan autonomously within four weeks after the presentation. During Task 1, an experienced clinician was present in the clinical training room to support the students when questions or problems arose during scanning. The duration of the intraoral scan was measured with a timer (mm:ss) running from the beginning of the capturing operation until the ATP pressed “save scan”.

### 2.5. Task 2: Computer-Aided Design (CAD)

The students performed the CAD of the bite splints (Dental Cad Designer, see Table 1) after a 45-minute seminar in a group of a maximum of ten students. The students were handed an illustrated hardcopy with a step-by-step guide and an instruction video (Appendix A). They were able to consult the study’s supervisor for advice on CAD design. The net process time of the CAD task was measured with a clock from the beginning of the task (sitting at the CAD-PC with software ready to import the scan data) to the completion of the design for further processing (save and close), as reported by the ATP.

### 2.6. Computer-Aided Manufacturing

Computer-aided manufacturing (CAM) was based on digital light processing with an acrylic resin (see Table 1). The study’s supervisor aligned the splints on the building platform and positioned the support structures to ensure reliable printing dimensions [46], including postprocessing with isopropanol (99%) in ultrasonic washing and light-curing (see Table 1).

### 2.7. Tasks 3 and 4: Finishing and Insertion

The participants had to remove the support structures of the splints, smoothen sharp edges, and polish in two steps (pumice slurry and polishing paste). Due to a limited number of workstations and polishing units, time was not recorded for the finishing step in order to avoid false records. Prior to insertions, the splints were disinfected for 2 min (OmniSept IMP, Omnident, Rodgau, Germany) and rinsed under flowing water. The ATP had to adapt the splint to achieve a proper fit with an equilibrated static and dynamic occlusion. It was the PTP’s discretion to wear the splint at night and have a follow-up examination. The duration of the insertion session measured by the ATP started at the first insertion of the splint into the PTP and ended with their feedback on occlusal comfort or the failure of the splint. This included documentation of the initial and final occlusal contacts marked by the occlusal foil possessing 12 µm thickness.

### 2.8. Data Acquisition

With enrollment, participants gained an individual pseudonym (ID). As an ATP, they were assigned to indicate the ID of their PTP in the evaluation forms. The questionnaires were rendered to the participants right before a task, and the forms were filled in immediately after they had completed their task.

In the case of VAS, the distance between the left pole and the mark was measured in mm, and it was noted as an integer value. The responses from all questionnaires submitted were entered into a data table and further processed statistically (JMP software package, 15.2, SAS Corp., Cary, NC, USA).

### 2.9. Data Validation and Statistical Analysis

Questionnaires were only allowed for evaluations in the case of bijective pseudonymizations. If no ID was provided, then the questionnaire was not evaluated. If the same ID was entered several times, then the questionnaire with the earlier ID was evaluated, and the other questionnaires with the same ID were discarded. Data from the VAS are depicted with distributions and described with mean values and standard deviations. For descriptive statistics, the distribution of VAS answers per item is grouped by relative frequencies within each third of the scale (0–33.3–66.6–100). In order to examine longitudinal stability of the items over the four cohorts, cumulative distribution function (CDF) and least-squares mean values as well as multiple factor analysis (Steel–Dwass all pairs) and restricted maximum likelihood (REML) estimations were applied. Relevant deviations were accepted whenever the 95% confidence intervals revealed no overlaps.

## 3. Results

All invited students (100%) signed the agreement, and 82 students (94%) participated in the study, ranging from 78 to 48 participants between the tasks. (Figure 2).

Due to incorrect or inconsistent pseudonymization, 40 of the 432 submitted questionnaires (9%) had to be discarded (Appendix A). Furthermore, missing values were also present in single questions of correctly pseudonymized questionnaires resulting in differing *N* per question. The data of the feedback VAS values can be found in Appendix A, and data from single- and multiple-choice items regarding each task are in Appendix A. The durations to perform the major activities within each task are provided in Table 2.

### 3.1. Feedback from Task 1: Intraoral Scan

Following Figure 3, most ATPs reported an easy and “fast” handling of the IOS. About two-thirds of the participants (69%) needed one attempt to perform the scan (Appendix A), while half of the participants (55%) paused the scan three to five times during a single jaw scan. Most participants said that they felt adequately prepared for the intraoral scanning, but 42% reported having difficulty when using the intraoral scanner.

The most frequent problem was exceeding the maximum recommended number of 1500 images during the scan (23%). The passive tandem partners (PTPs) perceived the scanning intervention as rather comfortable (95% VAS < 66) and swift (60% VAS < 33).

### 3.2. Feedback from Task 2: Computer-Aided Design

The digital design of the occlusal splint was reported as challenging by most students (61%) and, as shown among others in Figure 4, perceived as predominantly long-winded (80% VAS > 33; 50% VAS > 66). Two out of three participants rated the demonstration and the manual as sufficient (70% VAS < 33) for providing guidance for CAD design. Two-thirds of the students (*N* = 70) completed the design on the first attempt and, all in all, reported requiring no (33%) or slight (49%) support. Problems in CAD design were reported by 46% of the participants: design request (*N* = 17), system crash (*N* = 5), software malfunction (*N* = 9), IO scan captured occlusion (*N* = 2), bite registration, and insufficient hardware performance (each *N* = 1). Nevertheless, 48% of participants indicated that they would feel capable of designing splints on their own (VAS < 33).

### 3.3. Feedback from Task 3: Finishing

Following Figure 5, manually machining the additively manufactured splint material was rated by over 60% as “simple” (VAS < 33), with usual handling accompanied by “low efforts” in polishing (aside from five polishing problems), leading to a primarily good final result (VAS < 33) in 7 out of 10 cases. Problems occurred in hand with shortcomings in CAD design (*N* = 4), problems with the fit (*N* = 9), one visible deformation, and doubtful rating of mechanical stability (*N* = 6).

### 3.4. Feedback from Task 4: Insertion

In summary, 11 out of 84 (13%) splints failed mechanically within the workflow: One splint could not be printed (WS17-18) due to inadequate design. Seven fractures or cracks were documented when finishing 84 printed splints (8%). Three cracks and fractures appeared during insertion. Two cracks or fractures were documented in 14 splints of the WS18-19 and four each in SS18 (*N* = 21) and WS17-18 (*N* = 21), whereas no cracks or fractures were documented in SS19.

The initial fits of the documented splints (*N* = 61) were rated inconsistently by the ATPs: 68.5% of the participants described the initial fit as clinically acceptable. The retention of the splint against detachments from the final position was predominantly rated as clinically acceptable (62.7%), followed by a too-tight hold (23.7%) and a too-loose hold (13.6%).

After chairside adjustments with rotating instruments, three out of four splints with an initially clinically unacceptable fit (*N* = 16) could be fitted, resulting in four occlusal splints (7%) not achieving a clinically acceptable fit at the end of the session (Appendix A).

The number of initial static contacts varied between 1 and 18 contact points (median: 7). With an average of 49 min for adjusting an equilibrated occlusal surface (Table 2), about one-third of the participants each indicated medium (38%), low (36%), or high (26%) effort. Five participants (8%) were unable to achieve an equilibrated bite plane. In the end, nine out of ten splints produced were reported as successfully inserted, and the final result of the splints was predominantly rated by the ATPs as good (66% VAS < 33) and seven splints as failed (very poor).

With splint insertion, the PTPs reported an initial feeling of tension (53%) up until clamping (26%) as well as the tilting of the splint (16%). The feeling of wearing the occlusal splint was rated as most comfortable (Appendix A).

### 3.5. Reflection of the Workflow/General Preferences

The overall workflow with scanning, designing, finishing, and inserting the splints (*N* = 58) was predominantly rated very good to satisfactory (79%) by most participants (Appendix A). Concerning the training for dental professions, the participants evaluated the practice and implementation of the intraoral scan in the curriculum as helpful for later daily practice (Appendix A).

Barely fewer than half of the participants (47%) preferred the intraoral scan over conventional impression-taking, whereas one-third (32%) indicated both procedures as equivalent, and every fifth student (21%) would further prefer the conventional method. The participants generally preferred the digital fabrication of occlusal splints in undergraduate training, but did not express a clear preference for a particular method.

### 3.6. Longitudinal Observation of the Item Reproducibility

The feedback for items within the tasks and their durations were widely comparable over the four cohorts between October 2017 and July 2019 (Figure 6, Appendix A). A statistically significant difference (Steel-Dwass All Pairs, *p* < 0.01) was detected for the scan time of the ATP in SS19.

The WS18-19 showed a statistically significant (REML) shorter measured duration for the digital construction of the splint compared to SS18 (*p* < 0.01) and SS19 (*p* < 0.01). However, the perceived duration for the splint’s design during the semester was nearly equivalent (*p* > 0.3). The time required to fit and equilibrate the splint in Task 4 shows very similar values for all semesters and no significant deviations (*p* > 0.7).

### 3.7. Control for Individual Resources and Preferences

Eight participants reported a pre-existent experience with 3D and/or CAD software. They reported lower durations for the intraoral scan (9′, SD = 4′ vs. 19′, SD = 10′; *p* < 0.01) and fewer problems in the design task (13% vs. 53%), which correlated with significantly (*p* < 0.01) higher timely performances (1 h21′, SD = 30′ vs. 2 h27′, SD = 1 h12′) when designing. Interestingly, the 16 students who preferred the conventional impression method stated lower efforts in fitting the splint (*p* < 0.01) and reported a better fit after insertion (*p* = 0.06) (Appendix A).

## 4. Discussion

The current literature about IOS-based full digital workflows in undergraduate dental education addresses only the design and/or fabrication of single-tooth restorations or fixed dental protheses [20,44,45], but neither bite guards nor removable dental prostheses are addressed.

This implies that today’s graduating dentists have not had continuous contact or training with digital workflows in dentistry aside from an established system, such as market leader brand CEREC (Dentsply Sirona Comp.) [19,20,47,48]. Nevertheless, dental schools must contribute to the transformation in oral health care. In addition, the implementation of digital dentistry in university teaching is becoming increasingly mandatory [22]. Dental schools must take action, and a question may arise with respect to whether the extensive intraoral scanning of resin teeth on mannequin head-mounted models (combined with the fabrication of restorations) [20] is a prerequisite for applications in the patient. Such mannequin settings cannot simulate real-life conditions, as there are limitations such as salivation, gag reflex, and movement restrictions due to anatomical structures, and thus the precision results are not transferable [49].

The “analogue-trained” students were in their fourth year of dentistry education and had one year of patient care experience; thus, they can only be considered as novices with respect to digital dentistry. The approach of training them within a full digital workflow can be considered as a worst-case scenario in the process chain of intraoral scans in addition to using CAD to obtain a CAM workpiece without any casts or control except the fabricated splint itself. Therefore, only the trained handling of the additive manufactured splint (finishing and insertion) was their asset. Furthermore, the current intervention did not provide feedback for students during interim stages. Thus, problems in the constitutive work steps (such as splint fractures due to insufficient design) could not be recognized or prevented. This is a strength of the study, enabling the teacher to learn from the pitfalls of students as well as for students to learn from their own mistakes. In particular, the students had the opportunity to evaluate themselves during all steps by inserting a self-designed splint on a self-made intraoral scan [50]. However, a significant limitation of this study is the absence of validated questionnaires for comprehending attitudes and experiences in novel and complex processes. Therefore, a setup comprising questions and questionnaires should be developed for such types of studies in order to compare different studies. The results and experiences from our study may support that purpose.

Above all, the clinical setting chosen within this study validates the hypothesis that analogue-trained students can convey their manual capabilities successfully with respect to IOS handling if they are taught theoretically as well as with practical demonstrations that are passively supervised during performances.

As shown in the data, the integration of digital workflows into education is not only costly with respect to investments in equipment (see below), but it is also costly with respect to teaching operations and time when it comes to CAD training. Teaching in small groups (as applied in this study) has proven to be a successful method that is very well received by students [23]. However, it increases personnel costs for integrating new digital technologies into the curriculum [42,51]. Personnel expenses might be diminished by the use of asynchronous video tutorials and detailed written instructions that allow independent learning when using complex CAD software [48], as our data show.

In the context of teaching bite guards, 3D printing as an additive process offers numerous advantages over subtractive manufacturing. On one hand, the purchase of 3D printers (established devices from EUR 1500 to EUR 4000) is considerably cheaper than CNC milling machines (established devices from EUR 15,000 to EUR 200,000). At the same time, the costs of fabrication per workpiece are only a fraction of the total manufacturing cost, as considerably less waste and residual materials can be reused [52,53]. Even higher technical and personal efforts are required when designing supports and during postprocessing, which can be seen as a disadvantage. Unfortunately, the fracture strength of additively manufactured bite splints is reported to be lower than that of subtractively manufactured bite splints from industrial PMMA [54,55,56,57,58]. Aside from the good potential reparability of these materials [59], however, a material with a lower fracture strength may at the same time fail more easily in terms of cracks, fragments, or fracture. Particularly in the context of education, additive manufacturing offers cost-effective simultaneous production of several bite splints. In this manner, production can be coordinated and aimed toward the maximum utilization of the construction space while at the same time keeping the financial burden low.

Although more detailed research is needed, this study suggests that prior experience with CAD may influence an individual’s ability with respect to handling intraoral scanners. Therefore, universities should also consider the integration of intraoral scanning and CAD teaching into their early curricula and in hand with analog teaching activities [60]. Multiple applications—especially of intraoral scanning—also lead to a teaching effect regardless of prior experience, and these applications assist in the education of inexperienced users with respect to handling CAD designs in a short period of time [61]. Therefore, from an ethical point of view, the PTP experience also suggests that integrating a full digital workflow can take place in actual patient treatments by “digital inexperienced” students [62,63,64,65]. The insertion and fitting of a bite splint allows the use of CAD/CAM-manufactured dental workpieces in interactive dental exercises.

## 5. Conclusions

This study shows the positive view dental students have toward digital dentistry. The competencies from analog workflows allowed undergraduate students to exhibit an overall good performance with respect to utilizing complete digital workflows. Within the limits of this study, it can be shown that even digitally inexperienced—but dental-experienced—operators can prepare dimensional sufficient full-arch IO scans and fabricate clinically acceptable occlusal splints in a fully digital workflow with the use of a 3D printer.

From the educators’ side, it must be highlighted that learning CAD is the most time-consuming activity with a steep learning curve. This may be offset by facilities that allow regular and supervised CAD training, e.g., with a design computer pool on hand with instructional software.

## Figures and Tables

**Figure 1 dentistry-11-00005-f001:**
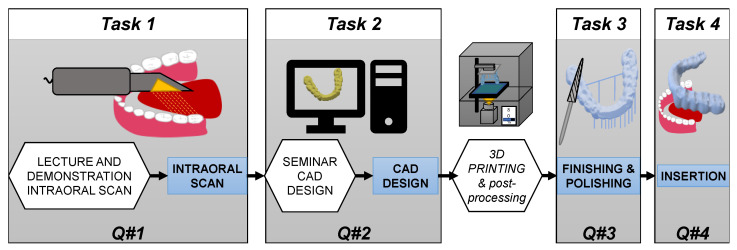
Study procedures: After instructions and demonstrations by the study supervisor (white hexagons), the active tandem partner (ATP) performed the intraoral scan on the passive tandem partner (PTP), which was followed by the CAD design, the lab finishing, and the insertion of the splint to the PTP. The study’s supervisor fabricated the splints via 3D printing and also handled postprocessing operations for the students.

**Figure 2 dentistry-11-00005-f002:**
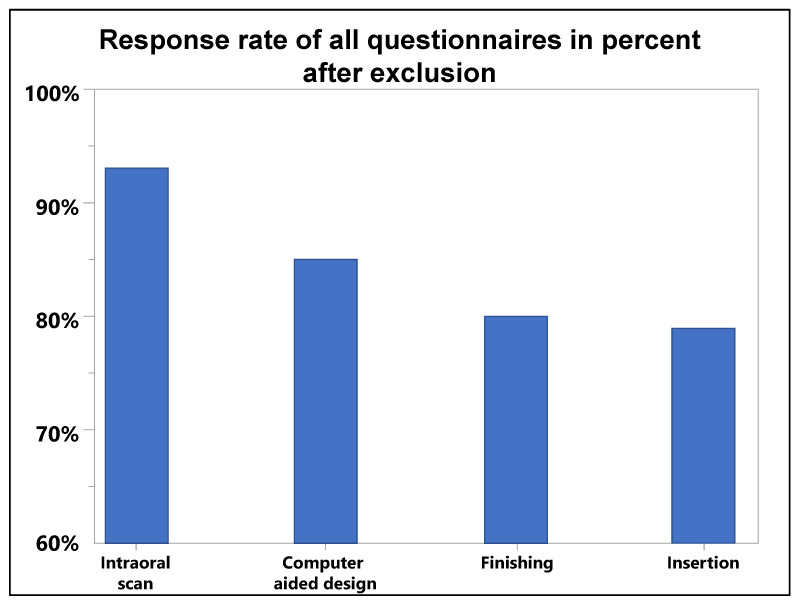
Response rate of questionnaires (*y*-axis, 100% = all enrolled participants) from the four tasks (*x*-axis) handled in all cohorts.

**Figure 3 dentistry-11-00005-f003:**
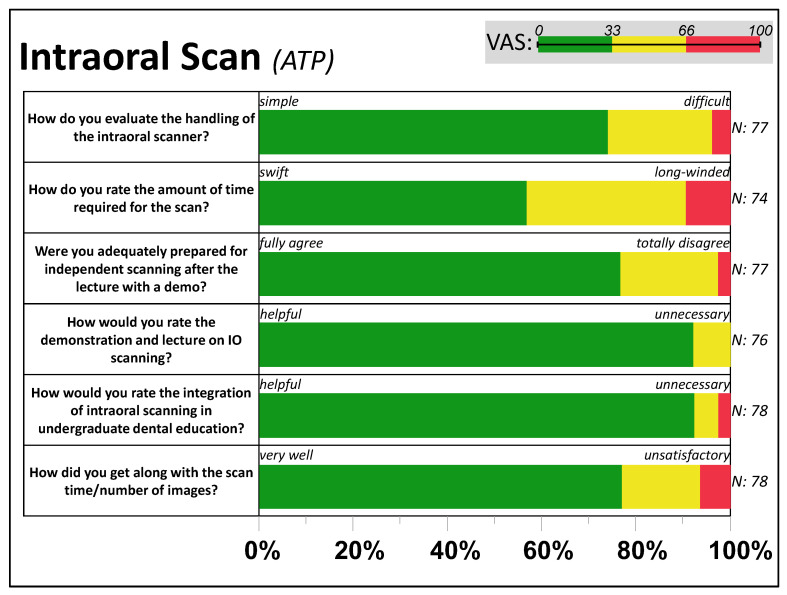
Feedback about intraoral scanning. The distribution of VAS answers per item, grouped by relative frequency within each third (0–33, 33–66, and 66–100) of the active tandem partners (ATP).

**Figure 4 dentistry-11-00005-f004:**
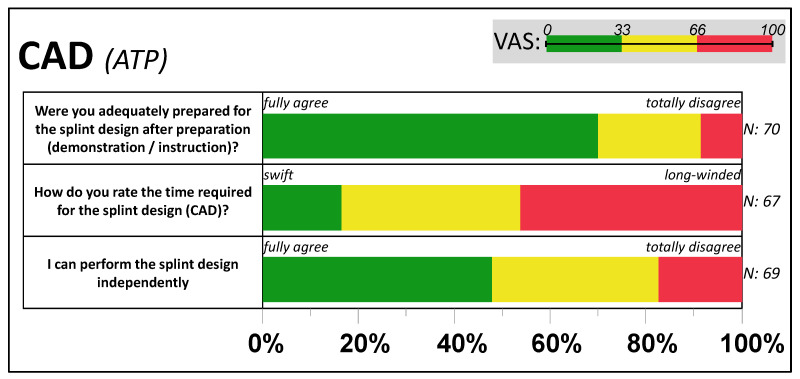
Feedback from computer-aided design (CAD) of the bite splint. The distribution of VAS answers per item, grouped by relative frequency within each third (0–33, 33–66, and 66–100) of the active tandem partners (ATP).

**Figure 5 dentistry-11-00005-f005:**
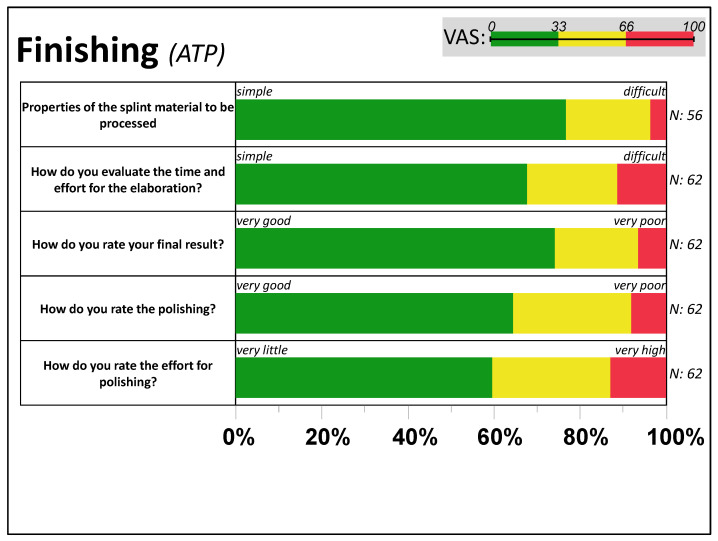
Finishing the additive manufactured bite splint. The distribution of VAS answers per item, grouped by relative frequency within each third (0–33, 33–66, and 66–100) of the active tandem partners (ATP).

**Figure 6 dentistry-11-00005-f006:**
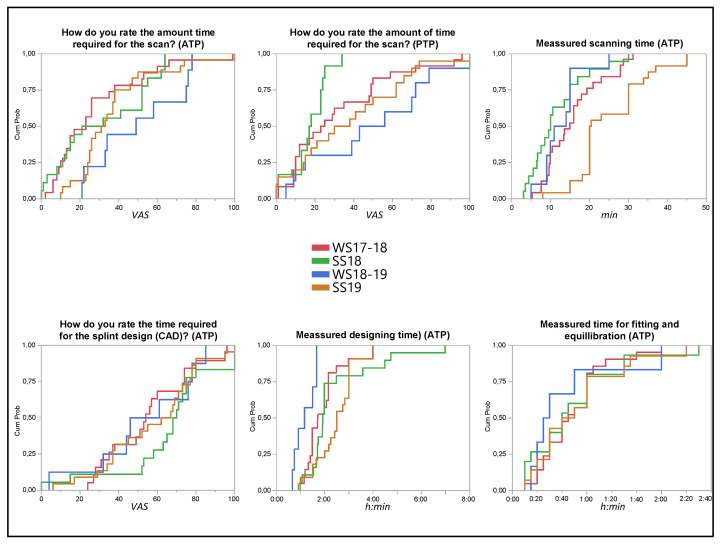
Responses to time-relevant items in each cohort. The CDF plots provide with each line (semester separated by color and legend in the middle) the mean values of the VAS answers (x-value; 0 = swift; 100 = extensive) and duration (x-value in h:mm) in the participants of the semester cohort. This allows a comparison of the cumulative probability (*y*-axis, sum of all participants in percent) of the responses of all participants (active and passive tandem partners ATP/PTP).

**Table 1 dentistry-11-00005-t001:** Devices and materials used to support the digital workflow performed by the students.

WORKFLOW	Company, City, Country	Technology	Specification, Software Version
Intraoral scan	3Shape, Kopenhagen, Denmark	Ultrafast optical sectioning	Trios 3 color, Pod Version, Ver: Trios 2015-1
CAD	exocad, Darmstadt, Germany	Dental CAD Designer with Bite Splint Module	DentalCAD 2016.10, Ver: Valetta 2.2, Matera 2.3
CAM prestage	Autodesk, California, USA	Slicing software for additive manufacturing	Netfabb Premium 2018 Netfabb Premium 2019
Additive manufacturing	W2P Engineering, Vienna, Austria	DLP, 385 nm, Flex-Vat	Solfex 650, Solflex 170
	VOCO, Cuxhaven, Germany	Resin: Dimethacrylat	V-Print Ortho

**Table 2 dentistry-11-00005-t002:** Durations of the major activities within the tasks as reported by the students (*N*) throughout the four cohorts.

Task	*N*	25th Percentile (h:m:s)	Median (h:m:s)	75th Percentile (h:m:s)	Mean (min)	SD	Min. (min)	Max. (min)
#1 intraoralscan	78	00:09:54	00:15:00	00:21:30	17.5	9.5	3	45
#2 splint design	69	01:30:00	02:00:00	02:38:00	132	65	40	420
#3 finishing	-	not evaluated						
#4 fitting of splint	56	00:25:00	00:40:00	01:00:00	49	33	10	150

## Data Availability

The data are provided in the manuscript as well as the Appendix A. Data beyond that granularity are privacy-sensitive and therefore are available from the corresponding author upon reasonable request.

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
