# Peer review of "Implementation of a Full Digital Workflow by 3D Printing Intraoral Splints Used in Dental Education: An Exploratory Observational Study with Respect to Students’ Experiences"

_dentistry, 2022, doi:10.3390/dj11010005_

Round 1

Reviewer 1 Report

In general terms, this paper is a useful piece of research however, the  English language style in parts is overly complex/convoluted and the paper would benefit from significant editing.

Abstract:

There is an inconsistency in the reporting of time.  The timing for the intra-oral scan should be reported in h (hours) and m (minutes) in the style reported in the splint design.

The conclusion should reference that this implementation is in an undergraduate setting.

Figure 2 is a line graph however, this is inappropriate given that these are discrete response rate categories.

Table 2 provides data on timing for 3 of the 4 tasks, task 3 is not reported and it is unclear why not.

Author Response

Point-by-point responses to reviewer 1:

We would like to acknowledge the editor and the reviewer for your time as well as both helpful and valuable comments. We have revised the manuscript accordingly. The changes are highlighted in the revised manuscript. We would also like to respond to the comments point-by-point as below.

R1: In general terms, this paper is a useful piece of research however, the  English language style in parts is overly complex/convoluted and the paper would benefit from significant editing.

[Reply]: Thank you very much for your positive comment encouraging us to revise our manuscript. The whole manuscript was edited by a professional authoring service.

R1:Abstract:

There is an inconsistency in the reporting of time.  The timing for the intra-oral scan should be reported in h (hours) and m (minutes) in the style reported in the splint design.

[Reply]: Thank you for the suggestion. We revised the manuscript as followed:

Line 25: The average time required to perform an intraoral scan was 17m5s, and all students were able to design a splint with an average of 2h38m.

R1: The conclusion should reference that this implementation is in an undergraduate setting.

[Reply]:The manuscript was edited as follow:

Line 29-30: The implementation of a digital workflow in undergraduate dental education is feasible and has acceptable clinical results.

Line 384: The competencies from analog workflows allowed undergraduate students to exhibit an overall good performance with respect to utilizing complete digital workflows.

R1: Figure 2 is a line graph however, this is inappropriate given that these are discrete response rate categories.

[Reply]: Thank you for pointing this out. Figure 2 has been replaced by a bar chart.

R1:Table 2 provides data on timing for 3 of the 4 tasks, task 3 is not reported and it is unclear why not.

[Reply]: Unfortunately, due to the limited capabilities at our facility, no meaningful timekeeping could be done for this step of the process (finishing and polishing) - therefore, there was no time measurement for task 3 and therefore no results. Due to the limited availability of workstations and polishing motors, waiting times may occur during regular semester operations, depending on the number of students, but may not necessarily correspond to the workpiece itself. We stated the circumstances in the material section and adjusted Table 2.

Line 154-156: Due to a limited number of workstations and polishing units, time was not recorded for the finishing step in order to avoid false records.

Reviewer 2 Report

Very interesting work from the point of view not only of dental education but also of the learning curve that every clinician must undertake with the introduction of increasingly new technologies.

The only thing I would like to point out is the fact that the survey has not been properly validated, since it seems to be a translation of a survey in another language.

I believe that this is a limitation that should be mentioned as such in the work.

Otherwise, the methodology is correct and the work is quite interesting.

Author Response

Point-by-point responses to reviewer 2

We would like to acknowledge the editor and the reviewer for your time as well as both helpful and valuable comments. We have revised the manuscript accordingly. The changes are highlighted in the revised manuscript. We would also like to respond to the comments point-by-point as below.

R2: Very interesting work from the point of view not only of dental education but also of the learning curve that every clinician must undertake with the introduction of increasingly new technologies.

[Reply]: Thank you very much for your positive comment encouraging us to revise our manuscript. In addition, the entire manuscript was edited by a professional authoring service after revision.

R2: The only thing I would like to point out is the fact that the survey has not been properly validated, since it seems to be a translation of a survey in another language. I believe that this is a limitation that should be mentioned as such in the work.

[Reply]: Many thanks for this advice. We have mentioned this aspect in detail within the discussion.

Line 338-3343: However, a significant limitation of this study is the absence of a validated questionnaires for comprehending attitudes and experiences in novel and complex processes. Therefore, a setup comprising questions/questionnaires should be developed for such types of stud-ies in order to compare different studies. The results and experiences from our study may support that purpose.

R2: Otherwise, the methodology is correct and the work is quite interesting.